cognition/behaviour/psychology

motion perception, VAR, slow-motion, decision bias, association football

**Author for correspondence:**
George Mather
e-mail: gmather@lincoln.ac.uk

# Is the perception of intent by association football officials influenced by video playback speed?

George Mather[1] and Simon Breivik[2]

[1]School of Psychology, University of Lincoln, Brayford Wharf East, Lincoln LN5 7AY, UK
[2]Professional Game Match Officials Limited, Brunel Building, 57 North Wharf Road, London W2 1HQ, UK

GM, 0000-0002-3102-4828

Recent research on motion perception indicates that when we view actions in slow-motion, the perceived degree of intent behind those actions can increase. Slow-motion replays are widely used in the checking and review of refereeing decisions by video assistant referees (VAR) in association football. To test whether the decisions of referees are subject to such a bias, 80 elite English professional football officials made decisions about 60 incidents recorded in professional European leagues (recorded as fouls, yellow-card offences or red-card offences by the on-field referee). Both real-time (1×) and slow-motion (0.25×) playback speeds were used. Participants had no prior knowledge of the incidents, playback speeds or disciplinary sanctions relating to each clip. Three judgements were made about each incident: extent of contact, degree of intent, and disciplinary sanction. Results showed an effect of playback speed on decision-making, but not a consistent bias due to slow-motion. Instead the distinction between yellow-card and red-card offences was clearer: Under slow-motion, yellow-card incidents were judged as less severe, and red-card incidents are judged as more severe, thus enhancing the distinction between these offences. These results are inconsistent with previous scientific reports that perceived intent is heightened by slow video playback speed.

## 1. Introduction

The speed of human actions carries information about the actor's emotions and intent [1–3]. Research also indicates that judgements of action speed and intent are liable to be biased in certain conditions. In a recent study of video replays [4], participants were asked to judge how much time the actor in a video of a sporting or criminal incident had to execute his actions, and whether they thought that the action was intentional. Results

showed significant increases both in time and in perceived intent using slow-motion replays compared to real-time replays, with serious implications for both sporting and criminal sanctions. Other recent studies [5,6] have found that extended viewing of slow-motion can alter one's perception of how fast a human action is performed. After viewing slow-motion for a while, real-time action can appear to be up to 30% faster, whereas slow-motion action can appear relatively normal. The latter shift may explain why actors viewed in slow-motion playback may appear to have more time to perform an (apparently normal) action: the action unfolds over a longer time period when viewed in slow-motion.

Slow-motion playback is used during video reviews that are widely employed in professional sport to advise officials during play. Video playback is already well-established in rugby league and cricket (from 2001), tennis (2002) and American Football (NFL; 2007). The video assistant referee (VAR) system was introduced into association football in late 2016 for the FIFA Club World Cup semi-final and has since been widely adopted internationally. According to the International Football Association Board (IFAB), VAR is now used by over 20 national football associations around the world and has had a major impact on decision-making by officials. VAR was introduced in the English Premier League in August 2019 and is used to aid decision-making on four potentially game-changing incidents:

Goals – was there an infringement that would rule out the award of a goal?
Penalty decisions – was the decision to award (or not award) a penalty correct?
Red-card decisions – was the decision to dismiss (or not dismiss) a player correct?
Mistaken identity – was a disciplinary sanction applied to the correct player?

According the Laws of the Game 2019/20 as set out by IFAB, judgements of intent or deliberation have relevance when penalizing hand-balls, brutality and violent conduct:

The law on hand-ball states: 'It is an offence if a player deliberately touches the ball with their hand/arm, including moving the hand/arm towards the ball.'

The definition of brutality in the law is 'An act which is savage, ruthless or deliberately violent'.

The definition of violent conduct includes actions 'when a player deliberately strikes someone'.

Slow-motion replays play a central role when a VAR checks or reviews the on-field decision of the referee. If, for example, a player handles the ball in the penalty area, VAR may be used to judge whether this act was intentional. Previous research indicates that the player may appear to have more time to perform the act when viewed in slow-motion, and the reviewer may therefore ascribe intent to the action and consequently sanction the player with a foul and yellow card. Such a decision is potentially game-changing, as it could lead to the dismissal of the player and the award of a penalty kick. Similarly, in slow-motion replay, the degree of intent behind physical contact between two players may appear to be higher, resulting in a player's dismissal for serious foul play or violent conduct.

IFAB protocols recommend caution in the use of slow-motion during VAR reviews:

'In general, slow motion replays should only be used for facts, e.g. position of offence/player, point of contact for physical offences and handball, ball out of play (including goal/no goal); normal speed should be used for the "intensity" of an offence or to decide if it was a handball offence' (IFAB Laws of the Game 2019/20)

However, the scientific evidence outlined earlier raises some serious issues about perceptual biases due to the use of slow-motion playback in reviews of refereeing decisions, with potentially serious consequences both for individual players, their teams, competition outcomes and, of course, for officials. There are two important areas of doubt surrounding slow-motion replays, which require further research. The first area of doubt concerns the scientific evidence supporting a change in perceived intent due to playback speed. The data in Caruso *et al.*'s [4] Experiment 2, which investigated the effect of playback speed in sport, is quite limited. So it would be premature to draw general scientific conclusions about slow-motion and impose restrictions on its use in VAR, on the basis of this data alone. The experiment used only a single clip in which a NFL player executes a prohibited 'helmet-to-helmet' tackle. Furthermore, the participants in that study were not experts in the sport, since they were recruited and participated online using the Amazon Mechanical Turk crowd-sourcing system. It remains to be seen whether other participants, particularly elite, specialist officials involved in professional association football would be liable to the same perceptual effects across a broader range of match incidents. A recent study of association football officials [7] found no difference in decision 'accuracy' using slow-motion rather than real-time replays, though a complex mixed ordinal regression model applied to the data in that study indicated that 'slow-motion clips are associated with increased odds for choosing a higher category on the decision scale' [7, p. 6]. So this latter study [7] obtained mixed results and furthermore did not measure the effect of slow-motion on perceived intent directly. Given both the scientific and practical implications of perceptual bias effects due to playback speed, it is important to resolve the issue.

The second area of doubt concerns scientific evidence relating to IFAB's VAR protocol, in which real-time and slow-motion playback speeds are used for different aspects of the decision: position of offence/point of contact (slow-motion) versus intensity/intent (real-time). Recent research shows that many judgements in perception are biased by very recent experience (serial order effects; [5,8,9]). Given that each VAR check involves viewing both real-time and slow-motion playback, can video reviewers avoid order effects and set aside what they see in slow-motion ('unsee' the slow-motion) when forming judgements of intensity and intent?

In the present study, 80 elite English professional officials were shown video clips of 60 incidents recorded from European football leagues (20 fouls, 20 yellow-card offences and 20 red-card offences, as judged by the on-field referee), at one of two view angles recorded by different cameras (wide or narrow) and played back at one of two speeds (real-time/1 × speed, or slow-motion/0.25 × speed). After viewing each incident twice, participants made three decisions about each incident: the heaviness of the contact and degree of intent evident in the incident, and the disciplinary sanction that should be applied. The playback speeds of the two views of each clip varied according to four speed conditions:

— Real-Real (RR): two real-time clips;
— Slow-Slow (SS): two slow-motion clips;
— Real-Slow (RS): first clip in real-time, second in slow-motion; and
— Slow-Real (SR): first clip in slow-motion, second in real-time.

The design aimed to resolve the following two research questions:

(1) **Slow-motion.** Does viewing slow-motion playback (SS) of a football incident influence expert decision-making, compared to viewing real-time (RR) playback?
(2) **Playback order**. Does viewing slow-motion playback of a football incident before real-time playback (SR) influence expert decision-making, compared to viewing slow-motion playback after real-time playback (RS)?

# 2. Material and methods

## 2.1. Participants

Eighty participants took part in the study (79 male, one female), drawn from the 100 professional officials forming Select Groups 1 and 2 of the Professional Match Game Officials Limited (PGMOL) panel, who officiate in all English Premier League and Championship matches. The participants included 34 members of Select Group 1 (Premier League) and 46 members of Select Group 2 (Championship). Thirty-one were referees and 49 were assistant referees. Data collection took place in November 2018 and January 2019. At that time, VAR was in use only on a trial basis in cup tournaments, so officials will have had very limited involvement in it.

## 2.2. Materials

The study employed video clips of 60 incidents that were recorded during televised matches in various European association football leagues (Saudi Pro, French Ligue 1, German Bundesliga, Turkish Super Lig, Dutch Eredivisie, Belgian First Division, Italian Serie A, Portugese Primeira Liga) by Hawk-Eye Innovations, who supply footage for all VAR reviews. Incidents were restricted to these leagues in order to ensure that English participants were not familiar with them. The 60 incidents were sub-divided by Hawk-Eye into three groups according to the severity of the incident (as indicated by the on-field referee's disciplinary decision): 20 fouls, 20 yellow-card offences and 20 red-card offences. Four versions of each incident were supplied by Hawk-Eye, making a total of 240 experimental stimuli: two of the four versions of each incident presented a wide-angle view as recorded during the match, and the other two presented a tight-angle view of the same incident, from a different perspective. Two playback speeds were supplied for each view angle; one in real-time (1.0×) and one in slow-motion (0.25×). These playback variations are similar to those that would be viewed during VAR reviews.

The clips were prepared so that the playback of each clip lasted 6 s. Therefore, the real-time version of each incident showed a longer period of on-field action than the slow-motion version. Hawk-Eye Innovations edited each video clip so that the slow-motion version was centred in time on the relevant part of the incident, such as the point of contact in a tackle. This presentation format was used because it is a realistic approximation to the clips that would actually be reviewed by officials; slow-motion

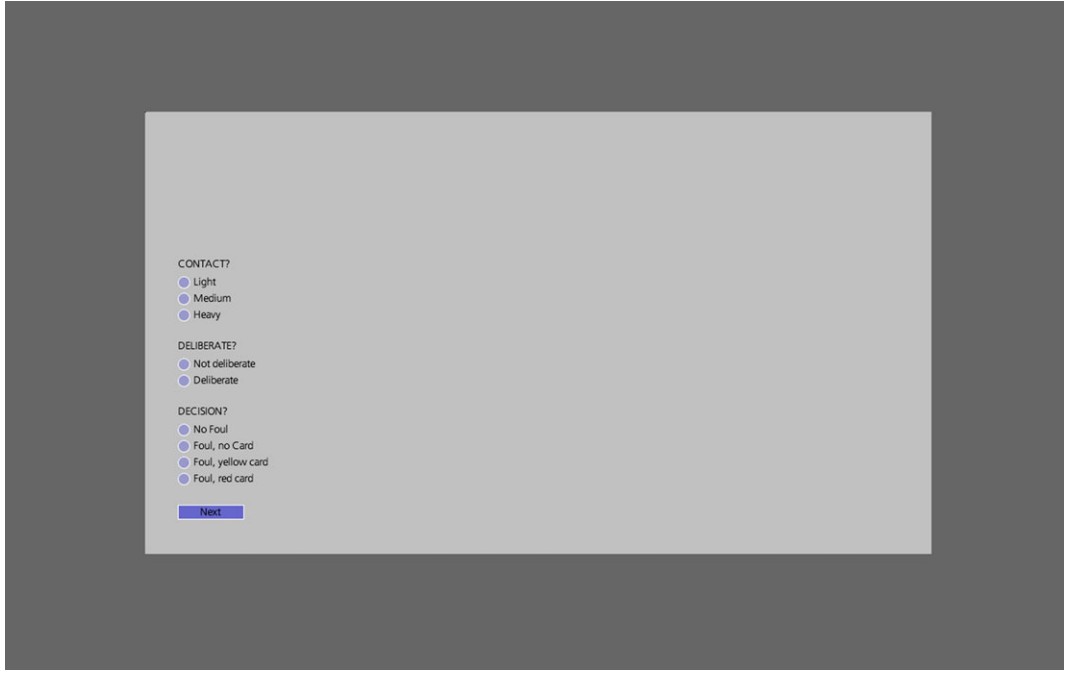

**Figure 1.** Screenshot of the display screen as it appeared to the participant. Each video clip was presented in the light-grey area of the screen, lasting 6 s before being removed and replaced by the light-grey background.

playback would normally focus on the contact phase of the incident, and real-time playback would allow the officials to view more of the build-up and aftermath of the incident. Each clip was provided at a resolution of $1920 \times 1080$ pixels at 50 fps, with a fixed duration of 6 s. Clips were re-scaled to a resolution of $960 \times 540$ pixels at 25 fps using Adobe Media Encoder, for use in the study. Data was collected using a custom-made Processing 3 Java application, running on Dell Windows 10 laptop computers.

## 2.3. Design and procedure

Clip view angle was a between-participants factor in the study design; half of the participants saw only wide-angle views, and the other half saw only tight-angle views. Playback speed and incident severity were within-participant factors, as follows.

Each participant took part in a single experimental session and saw each of the 60 incidents once in their session, one incident per trial, in order to avoid responses being contaminated by familiarity with specific incidents. Each trial involved viewing two successive clips of the relevant incident (a 'playback pair', separated by an interval of 0.5 s) before the participant was required to record a decision about it. The 60 trials were sub-divided into four blocks of 15, representing different playback pairs (RR, SS, RS, SR). The order of the four trial blocks varied across participants according to a randomized Latin square. The specific incidents shown in the different blocks also varied across participants to ensure that each incident appeared equally often in different playback speed conditions. Incident order within a block varied randomly across participants. Participants had no knowledge of the playback speeds or disciplinary sanctions relating to each clip.

Figure 1 shows a screenshot of the laptop screen as it appeared to the participant during the study. Videos appeared in the light-grey area of the screen. During the experiment, three multiple-choice questions were presented on the left-hand side of the screen, as follows:

— CONTACT?
— Light
— Medium
— Heavy
— DELIBERATE?
— Not deliberate
— Deliberate
— DECISION?

— No foul
— Foul, no card
— Foul, yellow card
— Foul, red card

The participant was instructed to watch the playback pair presented in each trial and then use the computer's trackpad to answer the three questions. Their instructions were as follows:

How do you rate the severity of the contact in the clip—light, medium, or heavy?
Do you judge that the contact was not a deliberate act by a player, or was a deliberate act?
What would your disciplinary decision be—no offence, offence but no card, yellow-card offence or red-card offence?

Once the participant had made their selections, he/she clicked the Next button to initiate the next trial. This process continued until all 60 trials had been presented. The participant's responses were stored on the PC in a CSV file. No information allowing identification of the participant was stored on the PC. Data were collected during two training days at St George's Park National Football Centre, Burton-upon-Trent, UK. Participants took part in groups of six to eight in a large meeting room, using a suite of laptop PCs spread across two large circular conference tables. Participants could not see each other's screen and completed the experiment at their own pace without discussion (each participant saw a unique combination of clips and conditions as dictated by the Latin square design, in a different random trial order).

## 2.4. Data aggregation

The 80 participants saw different subsets of the 240 stimuli, with the constraint that each participant saw a given on-field incident only once in their experimental session. Rather than analyse the data using mixed-effects models as in [7], we performed analyses based on aggregated data. Specifically, we used aggregation *by-stimulus*: the score for each stimulus was calculated by aggregating the responses to that stimulus across all of the participants who were shown it. We decided to use aggregation *by-stimulus* because this form of aggregation is not subject to a positive bias in our study, unlike aggregation *by-participant* [10]. Our participants can be considered as effectively a fixed factor in the sense that they formed the majority of the available population of elite group officials in English professional football. Mixed-effects models are also inherently complex, and their interpretation is not straightforward [11].

Three scores were calculated for each stimulus as follows:

contact score—proportion of 'heavy contact' selections as this selection is relevant for the most serious red-card offences;
intent score—proportion of 'deliberate' selections; and
disciplinary score—the average disciplinary level selected, where: 'No foul' = 0; 'Foul' = 1; 'Yellow card' = 2; 'Red card' = 3.

The two research questions above were addressed for each incident in each condition by comparing the relevant pair of scores for that incident, as follows.

To answer the slow-motion question, we compared the score for the SS playback pair against the score for the RR playback pair. If slow-motion playback biases judgements as suggested previously [4,7], clips should be judged as more intentional and more severe in the SS playback pair than in the RR pair, so the (SS–RR) difference should be positive for all incidents.

To answer the playback order question, we compared the score for the SR playback pair against the score for the RS playback pair. If the first clip played back influences judgements more than the second (cannot be 'unseen'), then clips should be judged as more intentional and more severe in the SR playback pair than in the RS pair, so the (SR–RS) difference should again be positive for all incidents.

Data aggregation from the 14 400 responses in the CSV files was performed in Matlab® and cross-checked for correctness against equivalent aggregations in Microsoft Excel®. Statistical analyses and data plots were performed in Matlab®.

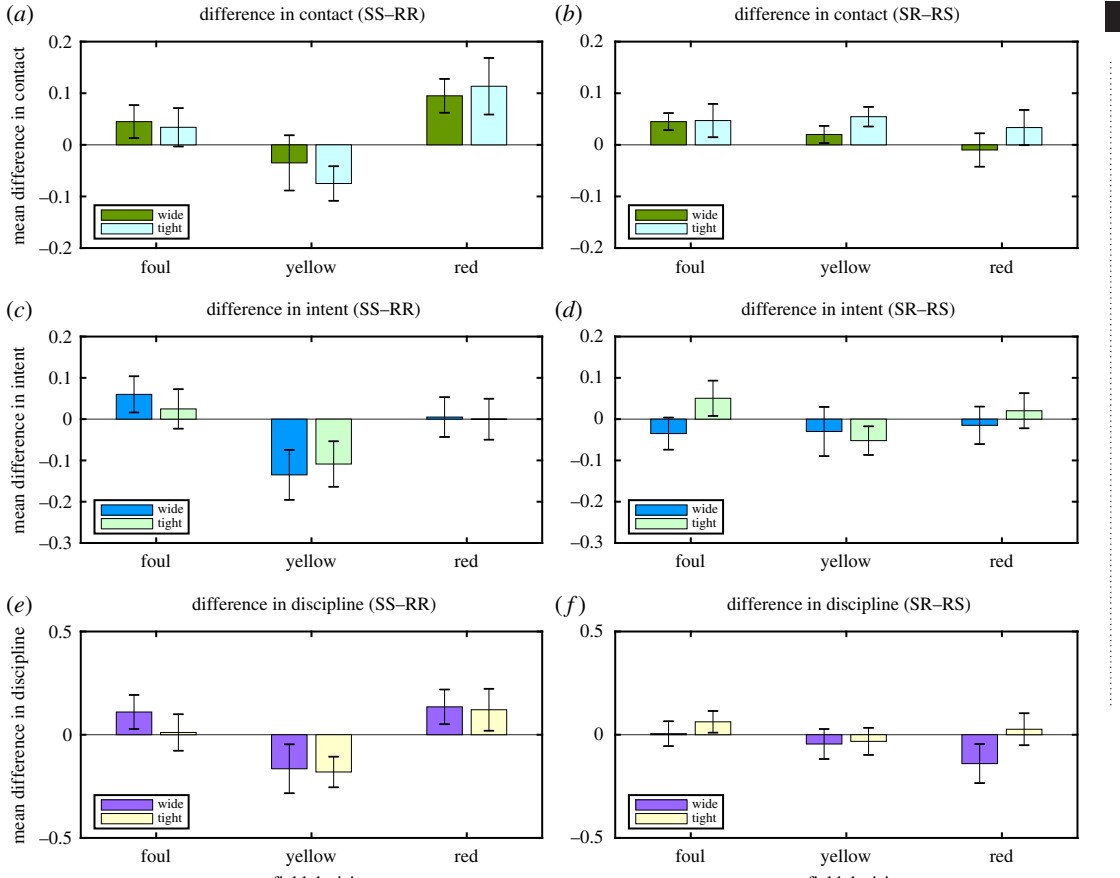

**Figure 2.** Summary of experimental results. The left-hand column of graphs plots differences between data in the SS and RR playback conditions, and the right-hand graphs plot differences between data in the SR and RS playback conditions. Graphs (a,b) show differences in contact heaviness scores, graphs (c,d) show differences in intent scores and graphs (e,f) show differences in disciplinary scores. Each graph plots three pairs of bars, corresponding to three levels of incident severity (fouls, yellow cards, red cards). Each pair of bars shows data separately for wide-angle views (left-hand bar) and tight-angle views (right-hand bar). Each bar shows the mean value across the twenty clips at each severity level, error bars represent ±1 s.e.m.

## 3. Results

The participants' judgements of contact heaviness, degree of intent and disciplinary sanction increased progressively in line with the severity of the incident as judged by the on-field referee. Data from relevant pairs of playback conditions were compared to assess the two research questions.

### 3.1. Slow-motion

The left-hand column of figure 2 shows the mean difference between data in the SS and the RR playback conditions. The top graph plots data on perceived contact heaviness, the middle graph plots data on perceived intent, and the bottom graph plots disciplinary scores. In each case, the difference is calculated on an incident-by-incident basis, and the graphs plot mean differences across the 20 incidents in each on-field severity level (foul/yellow/red; see Material and methods for details). So a positive mean difference would indicate that, on average, each incident was judged as more intense/intentional/severe in slow-motion than in real-time playback.

The graphs show that there is no consistent bias in judgements caused by slow-motion playback. Instead, the direction of the effect varies with the severity of the incident. For yellow-card incidents (middle bars in each graph), all three scores are actually lower for slow-motion playback than for real-time playback. For foul and red-card incidents, scores tend to be higher for slow-motion playback.

Three-factor mixed ANOVAs were performed on scores in the SS and RR conditions, treating the wide- and tight-angle views of the 60 incidents as six independent groups of 20 replicates (two

**Table 1.** Summary of ANOVAs to test the *slow-motion* effect. Rows in italics indicate statistically significant effects.

| SS vs RR | factor | F-ratio | d.f. | probability |
|---|---|---|---|---|
| contact | *ref* | *9.158* | *2, 114* | *0.001* |
| | angle | 0.011 | 1, 114 | 0.917 |
| | playback | 2.862 | 1, 114 | 0.093 |
| | ref × angle | 0.733 | 2, 114 | 0.483 |
| | *ref × playback* | *6.985* | *2, 114* | *0.001* |
| | angle × playback | 0.0954 | 1, 114 | 0.758 |
| | ref × angle × playback | 0.233 | 2, 114 | 0.793 |
| intent | *ref* | *20.832* | *2, 114* | *<0.001* |
| | angle | 0.075 | 1, 114 | 0.785 |
| | playback | 1.439 | 1, 114 | 0.233 |
| | ref × angle | 1.631 | 2, 114 | 0.200 |
| | *ref × playback* | *5.332* | *2, 114* | *0.006* |
| | angle × playback | 0.012 | 1, 114 | 0.913 |
| | ref × angle × playback | 0.172 | 2, 114 | 0.842 |
| discipline | *ref* | *19.728* | *2, 114* | *<0.001* |
| | angle | 0.015 | 1, 114 | 0.902 |
| | playback | 0.018 | 1, 114 | 0.893 |
| | ref × angle | 0.895 | 2, 114 | 0.411 |
| | *ref × playback* | *5.507* | *2, 114* | *0.005* |
| | angle × playback | 0.305 | 1, 114 | 0.582 |
| | ref × angle × playback | 0.131 | 2, 114 | 0.877 |

angles × 3 severity levels), and playback condition (SS versus RR) as repeated measures. Table 1 summarizes the results of the ANOVAs. The outcome is the same for all three decisions: there is a highly significant main effect of incident severity as judged by the on-field referee, because participants recorded progressively higher contact, intent and discipline scores as severity increased. There is a highly significant interaction between playback condition (SS versus RR) and incident severity (foul, yellow, red), confirming the statement above that the effect of slow-motion playback compared to real-time playback varies with incident severity. All other main effects and interactions were not significant.

In order to specifically test the effect of playback speed on all intent judgements, we combined the intent data across the two view angles and three levels of incident severity. Overall, the proportion of 'intentional' judgements was slightly lower (0.538) after slow-motion playback than after real-time playback (0.563), though this difference was not significant in a *t*-test ($t = 1.17$; d.f. = 119; $p$ (one-tail) = 0.12).

## 3.2. Playback order

The right-hand column of figure 2 shows the differences between data in the SR and the RS playback conditions for the three severity levels, calculated and displayed in the same way as in the left-hand column. The graphs show only relatively small and inconsistent differences in judgements between RS and SR in different conditions. Three-factor mixed ANOVAs were performed on the scores in the SR and RS conditions, treating the wide- and tight-angle views of the 60 incidents as six independent groups of 20 replicates (two angles × three severity levels), and playback condition (SR versus RS) as repeated measures. Table 2 summarizes the results of the ANOVAs. The outcome is mostly the same for all three decisions: there is a highly significant main effect of incident severity as judged by the on-field referee, but all other main effects and interactions were not significant except for a significant effect of playback speed on contact judgements (heavier contact in the SR condition than in the RS condition).

**Table 2.** Summary of ANOVAs to test the *playback order* effect. Rows in italics indicate statistically significant effects.

| SR vs RS | factor | F-ratio | d.f. | probability |
|---|---|---|---|---|
| contact | *ref* | *11.543* | *2, 114* | *<0.001* |
| | angle | 0.114 | 1, 114 | 0.737 |
| | *playback* | *8.221* | *1, 114* | *0.005* |
| | ref × angle | 0.102 | 2, 114 | 0.903 |
| | ref × playback | 0.866 | 2, 114 | 0.423 |
| | angle × playback | 1.459 | 1, 114 | 0.230 |
| | ref × angle × playback | 0.325 | 2, 114 | 0.723 |
| intent | *ref* | *22.442* | *2, 114* | *<0.001* |
| | angle | 0.483 | 1, 114 | 0.489 |
| | playback | 0.297 | 1, 114 | 0.587 |
| | ref × angle | 0.631 | 2, 114 | 0.534 |
| | ref × playback | 0.685 | 2, 114 | 0.506 |
| | angle × playback | 0.774 | 1, 114 | 0.381 |
| | ref × angle × playback | 0.690 | 2, 114 | 0.504 |
| discipline | *ref* | *18.715* | *2, 114* | *<0.001* |
| | angle | 0.038 | 1, 114 | 0.845 |
| | playback | 0.467 | 1, 114 | 0.496 |
| | ref × angle | 0.562 | 2, 114 | 0.572 |
| | ref × playback | 0.847 | 2, 114 | 0.432 |
| | angle × playback | 1.727 | 1, 114 | 0.192 |
| | ref × angle × playback | 0.577 | 2, 114 | 0.563 |

## 4. Discussion

These results clarify the two areas of doubt concerning the effect of playback speed on perceived movement and intent. The first research question asked whether viewing only slow-motion playback of an incident influences decision-making, compared to viewing only real-time playback. According to the data (see left-hand column of graphs in figure 2), we can answer this question in the affirmative: playback speed does influence decisions. However, the effect is not the one predicted on the basis of previous research. The prediction was that slow-motion playback would bias decisions in favour of higher intent, intensity and disciplinary sanctions, compared to real-time playback. On the other hand, the results showed that yellow-card incidents are judged as less intense, intentional and severe in slow-motion whereas red-card incidents are judged as more intense and severe. The ANOVA interaction effects are all highly significant (table 1). This result is not consistent with a previous report that slow-motion increases perceived intent [4]. Slow-motion playback increases the distinction between moderate and severe offences and thus may aid decision-making rather than hinder it. Although a longer period of on-field action is visible in the real-time clips, slow-motion may have allowed much closer scrutiny of the details of the contact phase of the incident and therefore greater clarity on the severity of the incident.

The second research question asked whether there is serial order effect: viewing slow-motion playback before real-time playback influences decision-making, compared to viewing slow-motion playback after real-time playback. The data in the right-hand column of graphs in figure 2 offer little support for the view that playback order matters. There are only very small differences between the two playback orders. Notice that the standard error bars of the difference scores mostly touch or cross the zero line, indicating that the differences are not significant. ANOVAs returned no significant effects apart from the effect of incident severity itself, and an effect of playback order on contact heaviness scores.

It is important to point out that the data presented above do not address the question of whether decisions are more or less likely to be 'accurate' using particular playback speeds. Indeed, there is no

ground truth of correctness or accuracy against which we can compare our data, because decisions of this kind are always based on the interpretation of the Laws of the Game by officials. With this proviso, after the data had been collected, we asked a panel of senior referees at PGMOL to view each incident at all of the available playback speeds and angles, in order to arrive at an agreed disciplinary decision. These decisions were compared to those made by the participants in the study, in each playback condition. The mean percentage of concordance between the participants and the panel across all clips seen by the participants in the slow-motion (SS) conditions was almost exactly the same as the mean concordance in the real-time (RR) conditions (55.4% and 55.0%, respectively). Previous studies [7,12] also report no difference in concordance levels between decisions in slow-motion and real-time conditions. Collapsing the data across playback speed conditions, but separately for foul, yellow-card and red-card decisions, mean concordance increased steadily with disciplinary severity (50% for fouls; 58% for yellow cards; 63% for red cards). Concordance percentages of around 60% may seem low, but they have been consistently reported in other studies of refereeing decisions [7,12] and may reflect the inherent difficulty of the task in terms of making a judgement based on applying the Laws of the Game to complex and variable real-life incidents, with very limited inspection time (just two 6 s views of each incident in our study).

# 5. Conclusion

In terms of the broader scientific implications for the study of perception, these results cast doubt on the view that slow-motion video can itself bias our perception of human actions to make them appear more intentional. Overall we found no significant difference between intent scores in the slow-motion and real-time conditions. It should be borne in mind, though, that our conclusions only apply to elite officials. The participants in the present study were highly trained and experienced in making very specific technical judgements. The question of whether other viewers such as TV football pundits or supporters are also immune to playback speed effects is open.

Regarding implications for the use of slow-motion playback during criminal cases or reviews in professional sport, our results show that decisions based on viewing slow-motion playback are not subject to a consistent bias due to playback speed itself. In sport, slow-motion replays may enhance the official's ability to discriminate between moderate and severe offences. The current VAR restriction on the use of slow-motion playback only for point-of-contact judgements (physical offences and handball) could therefore be relaxed. However, more research is required on the perceptual factors that bear on decision-making in VAR reviews.

Ethics. The study was performed in accordance with institutional guidelines and regulations. The experimental protocol was approved by the School of Psychology Ethics Committee, University of Lincoln, UK. All participants gave their informed consent to take part in the experiments.
Data accessibility. The raw data supporting this article have been uploaded as part of the electronic supplementary material. It is not possible to make actual video clips available due to copyright restrictions on their use imposed by Hawk-Eye Innovations.
Authors' contributions. Both authors designed the study and collected the data. G.M. analysed the data and prepared the first draft of the manuscript. Both authors reviewed the manuscript.
Competing interests. There are no conflicting or competing interests.
Funding. This research was made possible through support from the ESRC to G.M. (grant no. ES/K006088).
Acknowledgements. We are grateful to Hawk-Eye Innovations for their work in preparing the stimulus materials used in this study.

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
