## [Reviewer comments · Royal Society Open Science]

Review History

RSOS-192026.R0 (Original submission)

Review form: Reviewer 1

Is the manuscript scientifically sound in its present form?

Yes

Are the interpretations and conclusions justified by the results?

Yes

Is the language acceptable?

Yes

Do you have any ethical concerns with this paper?

No

Have you any concerns about statistical analyses in this paper?

No

Recommendation?

Accept as is

Comments to the Author(s)

File attached (Appendix A).

Review form: Reviewer 2 (Pieter Moors)

Is the manuscript scientifically sound in its present form?

No

Are the interpretations and conclusions justified by the results?

No

Is the language acceptable?

Yes

Do you have any ethical concerns with this paper?

No

Have you any concerns about statistical analyses in this paper?

Yes

Recommendation?

Major revision is needed (please make suggestions in comments)

Comments to the Author(s)

In this paper, the authors ask whether playback speed influences the perception of intent in association football incidents judged by video assistant referees. They assess perceived intent, extent of contact, and disciplinary sanction for a range of incidents in 80 elite officials. The results indicated that yellow card incidents are judged as less severe under slow motion whereas red card incidents are judged as more severe. They conclude that this observation is inconsistent with earlier research.

The authors build on recent research, identify some problems, and try to remedy these in this study. I think the question asked is timely and the methods to assess this question are appropriate. My main question pertain to the data-analysis. The remainder of my comments are very specific and are related to descriptions that were not clear to me or questions for clarification.

1. The authors aggregate the data by-stimulus, such that they acquire a score for each video clip in each condition. These scores are then visualized in Figure 2 and subjected to the repeated measures ANOVA reported in Tables 1 and 2. As the experiment consists of a pool of stimuli, and a pool of participants across both of whom we want to generalize, a mixed-effects model seems most appropriate to analyze these data (e.g., see Barr et al., 2013 or this blogpost for a very simple motivation: <https://debruine.github.io/posts/aggregating/>). That is, by-stimulus or by-participant analyses can, theoretically, be associated with increased false positive rates and simultaneously modeling both sources of variance can control for this. This is mainly the reason why we (Spitz et al., 2018) relied on a mixed-effects ordinal regression model. Indeed, given that this study relies on ordinal or binary measures, the authors could consider running mixed-effects ordinal (for contact and disciplinary score) or logistic regressions (for intent scores). I was about to run these models myself, given that the authors indicated that the data was available, but unfortunately such aggregated data can not be used for this type of analysis.

2. In the data statement, the authors mention that the “collated” dataset is available in the Supplementary Material. I had a look and either I missed something or either the authors simply

uploaded a tabular version of their Figure 2. I am sorry to be blunt about this but I do not consider this data sharing. Either the authors motivate why they do not want to share the raw data, or they simply upload the data. This kind of in-between “data set” very much feels like the authors needed to tick a box, and have no interest that an interested reader can verify the analyses that were reported in the paper (or, for whatever reason, might want to reanalyze this data set for other purposes).

3. I was wondering what “type” of officials were recruited in this study. Did they have experience as VAR? Were they mostly “regular” referees, or assistant referees?

4. I was confused by the description that “4 versions of each incident were supplied”. Did I read this correctly that every incident was either wide-angle or not and real-time or not? Or were there four versions of wide-angle and tight-angle (i.e., two of each)? The latter is how I understood it while reading.

5. The stimuli had a fixed duration of six seconds. How did the stimuli change in function of playback speed? Was there more information on the incident available in the real-time condition?

6. Why did the authors consider only the proportion “heavy contact” for the contact score?

7. I find it interesting that the authors acquired independent judgments on the disciplinary decision. In Spitz et al. we used this as the “ground truth”. I’m wondering what the concordance is between the severity of the video clips as coded by Hawk-Eye and as coded by the panel of senior referees.

8. I found it a bit weird that in both the Discussion and Conclusion new data and analyses were presented (especially in the Conclusion). Would it be more appropriate to move this to the Results section? Or at least restrict it to the Discussion section?

Signed,
Pieter Moors.

Decision letter (RSOS-192026.R0)

10-Feb-2020

Dear Dr Mather,

The editors assigned to your paper (“Is the perception of intent by association football officials influenced by video playback speed?”) have now received comments from reviewers. We would like you to revise your paper in accordance with the referee and Associate Editor suggestions which can be found below (not including confidential reports to the Editor). Please note this decision does not guarantee eventual acceptance.

Please submit a copy of your revised paper before 04-Mar-2020. Please note that the revision deadline will expire at 00.00am on this date. If we do not hear from you within this time then it will be assumed that the paper has been withdrawn. In exceptional circumstances, extensions may be possible if agreed with the Editorial Office in advance. We do not allow multiple rounds of revision so we urge you to make every effort to fully address all of the comments at this stage. If deemed necessary by the Editors, your manuscript will be sent back to one or more of the original reviewers for assessment. If the original reviewers are not available, we may invite new reviewers.

- Data accessibility

If you wish to submit your supporting data or code to Dryad (<http://datadryad.org/>), or modify your current submission to dryad, please use the following link:
<http://datadryad.org/submit?journalID=RSOS&manu=RSOS-192026>

- Competing interests

- Authors' contributions

AB carried out the molecular lab work, participated in data analysis, carried out sequence alignments, participated in the design of the study and drafted the manuscript; CD carried out the statistical analyses; EF collected field data; GH conceived of the study, designed the study,

coordinated the study and helped draft the manuscript. All authors gave final approval for publication.

- Acknowledgements

- Funding statement

Best regards,

on behalf of Dr Antonia Hamilton (Associate Editor) and Essi Viding (Subject Editor)
 openscience@royalsociety.org

Reviewers' Comments to Author:

Reviewer: 1

Comments to the Author(s)

File attached below

Reviewer: 2

Comments to the Author(s)

In this paper, the authors ask whether playback speed influences the perception of intent in association football incidents judged by video assistant referees. They assess perceived intent, extent of contact, and disciplinary sanction for a range of incidents in 80 elite officials. The results indicated that yellow card incidents are judged as less severe under slow motion whereas red card incidents are judged as more severe. They conclude that this observation is inconsistent with earlier research.

The authors build on recent research, identify some problems, and try to remedy these in this study. I think the question asked is timely and the methods to assess this question are appropriate. My main question pertain to the data-analysis. The remainder of my comments are very specific and are related to descriptions that were not clear to me or questions for clarification.

1. The authors aggregate the data by-stimulus, such that they acquire a score for each video clip in each condition. These scores are then visualized in Figure 2 and subjected to the repeated measures ANOVA reported in Tables 1 and 2. As the experiment consists of a pool of stimuli, and a pool of participants across both of whom we want to generalize, a mixed-effects model seems most appropriate to analyze these data (e.g., see Barr et al., 2013 or this blogpost for a very simple motivation: <https://debruine.github.io/posts/aggregating/>). That is, by-stimulus or by-participant analyses can, theoretically, be associated with increased false positive rates and simultaneously modeling both sources of variance can control for this. This is mainly the reason why we (Spitz et al., 2018) relied on a mixed-effects ordinal regression model. Indeed, given that this study relies on ordinal or binary measures, the authors could consider running mixed-effects

ordinal (for contact and disciplinary score) or logistic regressions (for intent scores). I was about to run these models myself, given that the authors indicated that the data was available, but unfortunately such aggregated data can not be used for this type of analysis.

2. In the data statement, the authors mention that the “collated” dataset is available in the Supplementary Material. I had a look and either I missed something or either the authors simply uploaded a tabular version of their Figure 2. I am sorry to be blunt about this but I do not consider this data sharing. Either the authors motivate why they do not want to share the raw data, or they simply upload the data. This kind of in-between “data set” very much feels like the authors needed to tick a box, and have no interest that an interested reader can verify the analyses that were reported in the paper (or, for whatever reason, might want to reanalyze this data set for other purposes).

3. I was wondering what “type” of officials were recruited in this study. Did they have experience as VAR? Were they mostly “regular” referees, or assistant referees?

4. I was confused by the description that “4 versions of each incident were supplied”. Did I read this correctly that every incident was either wide-angle or not and real-time or not? Or were there four versions of wide-angle and tight-angle (i.e., two of each)? The latter is how I understood it while reading.

5. The stimuli had a fixed duration of six seconds. How did the stimuli change in function of playback speed? Was there more information on the incident available in the real-time condition?

6. Why did the authors consider only the proportion “heavy contact” for the contact score?

7. I find it interesting that the authors acquired independent judgments on the disciplinary decision. In Spitz et al. we used this as the “ground truth”. I’m wondering what the concordance is between the severity of the video clips as coded by Hawk-Eye and as coded by the panel of senior referees.

8. I found it a bit weird that in both the Discussion and Conclusion new data and analyses were presented (especially in the Conclusion). Would it be more appropriate to move this to the Results section? Or at least restrict it to the Discussion section?

Signed,
Pieter Moors.

Author's Response to Decision Letter for (RSOS-192026.R0)

See Appendix B.

RSOS-192026.R1 (Revision)

Review form: Reviewer 2 (Pieter Moors)

Is the manuscript scientifically sound in its present form?

Yes

Are the interpretations and conclusions justified by the results?

Yes

Is the language acceptable?

Yes

Do you have any ethical concerns with this paper?

No

Have you any concerns about statistical analyses in this paper?

No

Recommendation?

Accept as is

Comments to the Author(s)

The reviewers have adequately addressed my comments.

Decision letter (RSOS-192026.R1)

03-Apr-2020

Dear Dr Mather,

It is a pleasure to accept your manuscript entitled "Is the perception of intent by association football officials influenced by video playback speed?" in its current form for publication in Royal Society Open Science. The comments of the reviewer(s) who reviewed your manuscript are included at the foot of this letter.

Kind regards,

Andrew Dunn

on behalf of Dr Antonia Hamilton (Associate Editor)
openscience@royalsociety.org

Reviewer comments to Author:

Reviewer: 2

Comments to the Author(s)

The reviewers have adequately addressed my comments.

Appendix A

Is the perception of intent by association football officials influenced by video playback speed?

Mather & Breivik

This paper purports to investigate whether referees' decisions are biased by viewing slow motion replays. Previous research has suggested that at slow speed actions may appear more intentional. Furthermore the order of viewing may alter perception; after viewing slow motion, real motion may appear faster.

To investigate the effects of slow motion replays in football the authors have carried out an impressively designed experiment, recruiting 80 professional referees and 60 video clips of infringements.

Each participant saw each incident twice; in some case the two presentations were in in real time (RR), sometimes both in slow motion (SS), sometimes (RS) and (SR). Three decisions were made: the severity, how deliberate it was, the appropriate disciplinary action.

Comparison of SS vs RR should give us an index of the effect of slow motion. The RS vsSR comparison tells us about possible order effects.

In summary this is a well-designed experiment asking very pertinent questions. The only disappointment comes in the results, scarcely the authors' fault. It would appear that contrary to previous research 'moderate' (yellow card) offences are judged *less* severe in slow motion. Red card offences are seen as heavier in slow motion but there is no effect on the appropriate discipline.

The authors conclude that previous research must be treated with caution in the light of these data. Absolutely. Further they suggest slow motion may allow referees to discriminate between yellow and red card decisions. Well, maybe. The truth is that we need a lot more research in this area to sort out what's going on, but this paper is a valuable addition to the literature as it is well- designed and executed.

Appendix B

Response to reviews

Reviewer 1

We thank the reviewer for their positive comments on the paper. We agree with the reviewer's final point that more research is required in this area, and have reinforced it in the revision (final sentence of the conclusion).

Reviewer 2

We thank the reviewer for their detailed comments on the paper. Our responses to their comments are listed below point-by-point.

1. The reviewer suggests that a linear mixed-effects model (LMEM) seems most appropriate to analyse the data, due to the theoretical possibility of an increased false-positive rate using data aggregation methods.

We agree that false-positives are a theoretical possibility using aggregation, and considered the issue carefully at the outset of the data analysis. However, we decided to use a form of data aggregation rather than a LMEM because we concluded that false-positives were not a problem in our method (our reasoning is summarised below). Furthermore, an aggregated analysis is relatively intelligible and transparent for the reader whereas LMEMs are complex and their interpretation is not straightforward, as noted by Barr et al. (2013, cited by the reviewer): The validity of a LMEM analysis also depends on the assumptions built into the specific model and test implemented.

In order to explain our reasoning in detail, we should outline the two ways in which our data could have been aggregated:

By-stimulus – The rating for each stimulus in each condition is calculated by combining the responses of the participants who were shown that stimulus. Stimuli are the unit of analysis.

By-participants – The rating for each participant in each condition is calculated by combining their responses to the stimuli in that condition. Participants are the unit of analysis.

We initially considered both kinds of aggregation, and decided that it was most appropriate to base the analysis on aggregation *by-stimulus*. As noted in Lisa DeBruine's blog (which the referee cites, and which we had read before deciding which form of aggregation to use) and in Judd et al. (2012), aggregation *by-participants* increases the chances of obtaining a false-positive difference between different stimulus conditions. We are primarily interested in the between-stimulus effect of playback speed, and aggregation *by-stimulus* would not increase the false-positive rate according to Lisa DeBruine and Judd et al. (2012). The participants in our experiment can also be considered as effectively a fixed factor in the sense that they formed 80% of the available population of elite group officials in English professional football.

We also concluded that our design is not vulnerable to a false-positive problem caused by chance differences between groups of stimuli in different playback speed conditions,

because the different groups of stimuli were so closely matched, and the differences due to playback speed reported in the graphs were all calculated on an incident-by-incident basis.

We have added a paragraph at the start of the Data Aggregation section to explain these points and to justify the use of analysis based on aggregation. We have also emphasised in the Conclusion that the results apply only to this group of officials.

2. We agree that the supplied dataset is not sufficiently detailed for a new analysis to be run from the raw data. An Excel file containing all the raw data has been prepared and will be uploaded. It lists the data in all 4800 trials of the experiment, each involving three participant responses.

3. The participants included 34 members of Select Group 1 (Premier League) and 46 members of Select Group 2 (Championship). 31 were referees and 49 were assistant referees. Data collection took place in November 2018 and January 2019. At that time VAR was in use only on a trial basis in cup tournaments, so officials will have had very limited involvement in it. This detail has been added to the Participants section.

4. There were indeed four versions of each on-field incident, two wide-angle and two tight-angle; in each pair, one version was real-time and the other was slow-motion. We have tried to make this more clear in the revised text in the Materials, Design and Procedure sections.

5. Yes, more information was available in the real-time version of each incident, because it showed a full 6 seconds of on-field action. In the slow-motion version provided by Hawk-Eye, only 1.5 seconds of on-field action was available. Hawk-Eye edited each video clip so that the slow-motion version was centred in time on the relevant part of the incident, such as the point of contact in a tackle. This presentation format was used because it is a realistic approximation to the clips that would actually be reviewed by officials; slow-motion playback would normally focus on the contact phase of the incident, and real-time playback would allow the officials to view more of the build-up and aftermath of the incident. We now make this clear in the Materials section, and comment on this difference in the Discussion.

6. Given that the most critical decisions in terms of their impact on the game are the most serious ones involving potential red-card offences, we decided that the most relevant aspect of the 'contact' judgement concerned 'heavy' versus 'light' contacts, rather than 'no contact' judgements. This is now stated in the Data Aggregation section.

7. The concordance between the Hawk-Eye/on-field severity scores and the panel scores was 55%, as reported in the paper.

8. The panel scores only became available after all of the analyses had been completed, and did not form part of the rationale for the design, so we decided to present them in the Discussion rather than somehow build the panel scores into the rationale retrospectively. We now make this point explicitly. We also moved the additional analysis mentioned in the Conclusion into the Results section.